# Human RBM3 protein is prone to form neuronal aggregates opposed by the proteasome

Suman Kumar, Tina Kleven and Rafal Ciosk*

## ABSTRACT

Maintenance of proteostasis is critical for neuronal functions, as the accumulation of misfolded or damaged proteins leads to neurodegeneration. Cooling is generally neuroprotective and is used in various clinical settings. However, how it impacts neuronal proteostasis remains unclear. In rodents, the neuroprotective effects of cold have been largely attributed to the cold-inducible RNA-binding motif protein 3 (RBM3). Here, studying the human RBM3 in cultured neurons subjected to profound hypothermia, we observed its cold-induced aggregation. These RBM3 aggregates are distinct from stress granules, occur specifically in differentiated neurons, and form also at physiological temperature upon proteasomal inhibition. Thus, in humans, RBM3 aggregation may be normally counteracted by the proteasome to maintain neuronal health. Exploring the natural variation between RBM3 proteins in hibernating versus non-hibernating mammals, we discuss how the aggregation could be prevented in animals with fluctuating body temperature. These findings are important for the understanding of RBM3 functions and neuronal proteostasis and have implications for medical treatments involving incidental and induced hypothermia.

KEY WORDS: RBM3, Aggregation, Proteostasis, Hypothermia, Hibernation, Neurodegeneration

## INTRODUCTION

Protein homeostasis, or proteostasis, maintains the functional integrity of the cellular proteome. Its decline can be manifested by the appearance of abnormal protein aggregates often associated with disease states. Neurons are particularly vulnerable to the deterioration of proteostasis and two major protein-degradation systems, the ubiquitin-proteasome system (UPS) and the autophagy-lysosomal system, ensure the removal of aggregated proteins. A failure in neuronal proteostasis can lead to the accumulation of pathological protein aggregates characteristic of neurodegenerative diseases such as Alzheimer's (AD) or Parkinson's (PD) (Lim and Yue, 2015).

Among factors opposing neurodegeneration, the RNA-binding motif protein 3 (RBM3) has received much attention. This protein plays various biological roles, including in neuronal homeostasis (Zhu et al., 2016). It contains an N-terminal RNA-binding domain (RRM) and a highly flexible C-terminal GR-rich region containing several RGG/RG motifs (Ciuzan et al., 2015; Thandapani et al., 2013). In neurons, RBM3 is upregulated upon hypothermia and is thought to stimulate neuronal plasticity and cognitive functions (Bastide et al., 2017; Ou et al., 2018; Peretti et al., 2015; Tong et al., 2013). Remarkably, the overproduction of RBM3 also at standard body temperature is sufficient to improve synaptic integrity and function in murine models of AD and prion disease (Peretti et al., 2015). However, whether the human protein similarly bolsters neuronal functions and preservation remains unclear.

While the human body maintains constant temperature, cooling is widely used in organ preservation. Also, since the first half of last century, therapeutic hypothermia has been used to facilitate the recovery from brain injury (Bernard and Buist, 2003). Cooling patients to 32-34°C is used routinely as part of the TTM (targeted temperature management) approach, applied to patients with spontaneously returning blood circulation after cardiac arrest (Gunn et al., 2017; Hunter and Ellender, 2015). The TTM facilitates preservation of energy reserves, inhibits inflammation and cell death, and stabilizes the blood-brain barrier (Yenari and Han, 2012). In a more radical approach, reducing core body temperature to around 10°C is used in EPR-CAT (emergency preservation and resuscitation for cardiac arrest from trauma) (Kutcher et al., 2016). This treatment protects the brain and other key organs of trauma victims but, in contrast to moderate cooling, much less is known about the underlying mechanisms.

This prompted us to examine RBM3 expression in human neurons subjected to profound hypothermia. In neurons subjected to 10°C (the target temperature in EPR-CAT), we observed the formation of cytoplasmic RBM3 aggregates. Intriguingly, these aggregates only form in differentiated neurons and can also be induced at standard body temperature (37°C) upon proteasomal inhibition. We propose that these RBM3 aggregates may be normally removed by the proteasome to maintain neuronal health, perhaps analogous to the removal of pathogenic aggregates associated with neurodegeneration. Finally, based on comparative analysis of RBM3 proteins across mammals, we hypothesize that animals experiencing hypothermia as part of their life cycle may have evolved aggregation-resistant RBM3 variants.

## RESULTS

### Acute hypothermia promotes the formation of RBM3 aggregates in differentiated neurons

To characterize the neuronal expression of human RBM3 under profound hypothermia, we employed the LUHMES (Lund human mesencephalic) cell line (Lotharius et al., 2002). Unlike many immortalized cells, LUHMES cells are diploid and have a stable karyotype. They are easily propagated as immature neuronal progenitors, thanks to tetracycline-controlled (Tet-off) expression of the retroviral element v-myc, but, upon the v-myc repression, will rapidly differentiate into dopaminergic-like neurons (L-neurons for short). This makes this cell line an excellent model for studying

Section for Biochemistry and Molecular Biology, Department of Biosciences, University of Oslo, Oslo 0316, Norway.

*Author for correspondence (rafal.ciosk@ibv.uio.no)

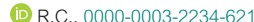 R.C., 0000-0003-2234-6216

neuronal health and degeneration, including in PD (Lalli et al., 2020; Lotharius et al., 2002; Scholz et al., 2011; Shah et al., 2016).

To monitor the effects of hypothermia, the L-neurons were exposed to different temperatures and analyzed as explained in each experiment (Fig. 1A). First, to monitor RBM3 protein levels, L-neurons were incubated at physiological (37°C) temperature, and either mild (30°C) or acute (10°C) hypothermia; note that the cooling procedure used in this study did not impair neuronal viability (Fig. S1A,B). Next, RBM3 protein was detected by western blotting using a specific antibody. In line with previous studies (Bastide et al., 2017; Ou et al., 2018; Tong et al., 2013), we observed that RBM3 levels were higher at 30°C than 37°C, but were similar between 10°C and 37°C (Fig. 1B,C). Next, we used immunocytochemistry to examine the cellular distribution of RBM3. At 37°C and 30°C, RBM3 predominantly localized throughout the nucleus, with some expression observed in the cytoplasm (Fig. 2A). At 10°C, in contrast, RBM3 was largely present in variably sized aggregates, some of which appeared to be away from the neuronal cell body (Fig. 2A). We did not observe such aggregation in undifferentiated LUHMES cells exposed to 10°C (Fig. S2A), suggesting that the aggregation of RBM3 upon acute hypothermia is specific to differentiated neurons. To better illustrate this behavior, we evaluated the numbers and size distribution of RBM3 aggregates at 37°C, 30°C, and 10°C. On the one hand, we observed increased numbers of the aggregates at 10°C compared with 37°C or 30°C (Fig. S2B). On the other hand, we noticed the emergence of particularly large aggregates (Fig. S2B). Since the smaller aggregates are most abundant, the mean sizes were comparable across the temperatures (Fig. S2B). Thus, to quantify the largest

aggregates, we introduced an arbitrary cutoff (0.5 μm³) and observed a significant enrichment of these aggregates at 10°C (Fig. 2B).

## Neuronal RBM3 aggregates form in the cytoplasm in cell bodies and neurites

The neuronal RBM3 is present in both the nucleus and the cytoplasm, and its RGG motifs appear to play a role in its distribution

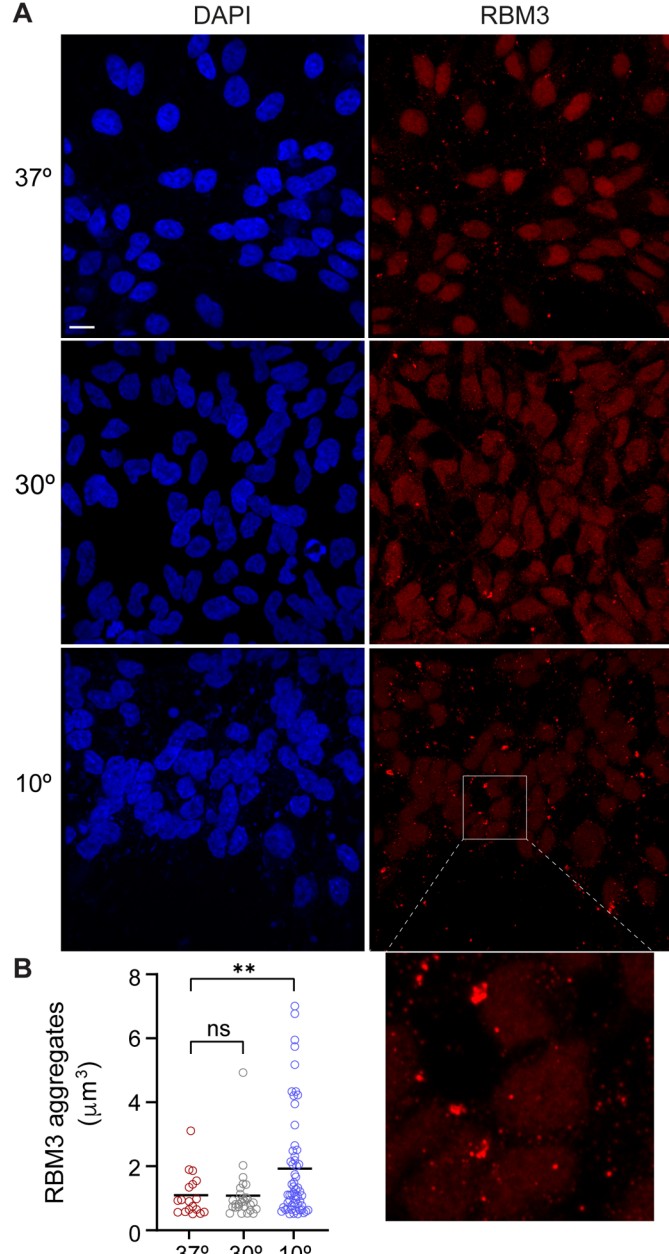

**Fig. 2. RBM3 forms aggregates in neurons subjected to acute hypothermia.** (A) Immunodetection of RBM3 at different temperatures. The L-neurons were incubated as indicated and stained with RBM3 antibodies and DAPI. The boxed area in the bottom-right panel is magnified below. Scale bar: 10 μm. (B) Quantification of RBM3 aggregates shown in A. The aggregates were scored per circa 300 cells/condition (37°C, 30°C, or 10°C) from three biological replicates (n=3), with the size cutoff of 0.5 μm³. The number of aggregates per cell was 0.06 at 37°C, 0.09 at 30°C, and 0.19 at 10°C. Welch's t-test was used for two-condition comparisons: 10°C versus 37°C, **P=0.003; 10°C versus 30°C, **P=0.003; 30°C versus 37°C, P=0.954 (ns).

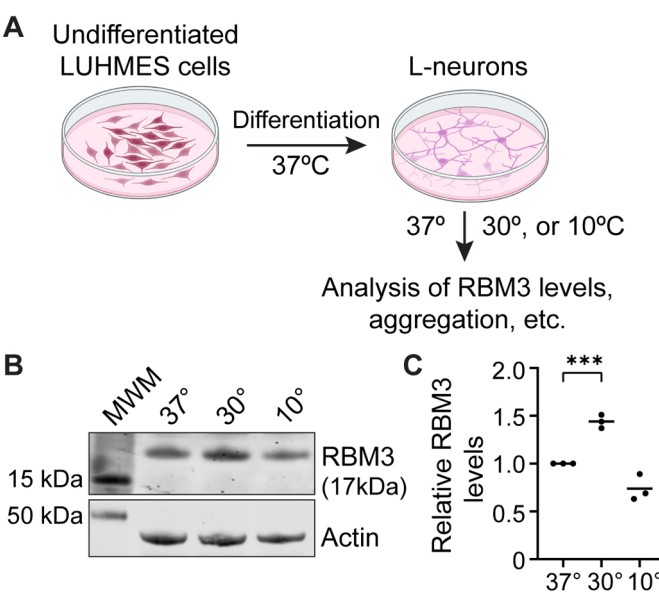

**Fig. 1. Mild but not deep cooling upregulates RBM3 in human neurons.** (A) Schematic of a representative experimental pipeline. First, LUHMES cells were differentiated into L-neurons over 5 days. Next, the neurons were either maintained at 37°C or incubated at lower temperatures: 30° for 16-24 h (moderate cooling) or 10°C for 16-24 h (deep cooling). Then, the neurons were analyzed depending on a specific experiment. (B) Comparing RBM3 protein levels at different temperatures. LUHMES cells were differentiated into L-neurons (as in A) and incubated at 37°C, 30°C, or 10°C. Shown is a western blot detecting RBM3 in whole lysates from L-neurons incubated as indicated. Actin was used as a loading control. (C) Quantification of RBM3 levels from three independent experiments (n=3), normalized to actin and shown relative to 37°C. Unpaired two-tailed t-test was used to calculate statistical significance. ***P=0.0004.

(Wang et al., 2023a). To examine the intracellular localization of RBM3 aggregates, we generated a pool of LUHMES cells stably expressing the plasma membrane marker LCK-mGreenLantern; the membrane-targeted GFP allows the visualization of the whole neuron including its processes (Benediktsson et al., 2005). Staining these neurons for RBM3, we observed that the RBM3 aggregates found away from the cell bodies co-localized with the LCK-mGreenLantern-labelled neurites (Fig. 3). Intriguingly, some of these neurite-associated aggregates appeared to be caught in the act of budding off (Fig. 3, inset). We additionally examined the aggregates within cell bodies and found that they were present in the cytoplasm rather than the nucleus (Fig. S3). Thus, by contrast to the physiological temperature at which RBM3 is mostly nuclear, acute hypothermia promotes the aggregation of RBM3 in the cytoplasm, both within cell bodies and neurites.

### RBM3 aggregates also form at a physiological temperature upon proteasomal inhibition

Many neuronal RBM3 aggregates are very large, and some may be budding off (Fig. 3). These characteristics are reminiscent of the recently identified large extracellular vesicles called exopheres, whose formation is enhanced upon inhibiting the proteasome (Melentijevic et al., 2017). These studies prompted us to test if the formation of RBM3 aggregates could be similarly induced by inhibiting the proteasomal function at the physiological temperature.

To test it, we treated L-neurons, continuously incubated at 37°C, with the proteasome inhibitor MG132. Following the exposure to MG132 (20 μM for 6 h), we observed the aggregation of RBM3 (Fig. 4A,B and Fig. S4A). Like in the cold, the MG132-induced aggregation at 37°C depended on neuronal differentiation (Fig. S2A). In hibernating animals, proteasomal degradation is thought to be globally suppressed, with the concomitant accumulation of ubiquitinated proteins (van Breukelen and Carey, 2002; Velickovska et al., 2005; Velickovska and van Breukelen, 2007). Thus, we globally examined protein ubiquitination in cold-treated L-neurons by western blotting. By contrast to hibernators, we observed no difference in the levels of ubiquitinated proteins between cells incubated at 37°C versus 10°C (Fig. S4B). Thus, either the proteasomal degradation continues in cold-treated cells, or cold globally inhibits protein ubiquitination. We also purified RBM3 from cold-treated L-neurons to examine its

potential ubiquitination but found no evidence for it (Fig. S4B). Thus, while the status of proteasomal degradation in cold-treated neurons remains unresolved, the ubiquitination of RBM3 does not appear to be necessary for the aggregation.

### RBM3 aggregates are distinct from stress granules

Cellular stress, including by severe cold, is known to induce the formation of stress granules (SGs) that are also implicated in neurodegeneration (Wolozin and Ivanov, 2019). The SGs contain various mRNAs and associated translation regulators such as the SG-nucleating protein G3BP1 (Tourriere et al., 2023). Since RBM3 has translation-related functions and was reported to localize to SGs (Si et al., 2020), we initially examined the relationship between RBM3 aggregates and G3BP1-labelled SGs in L-neurons. Expectedly, we observed a significant co-localization between RBM3 and G3BP1 in SGs induced by arsenite at 37°C (Fig. 5A,B). By contrast, the two proteins did not co-localize at 10°C (Fig. 5A,C). Moreover, we

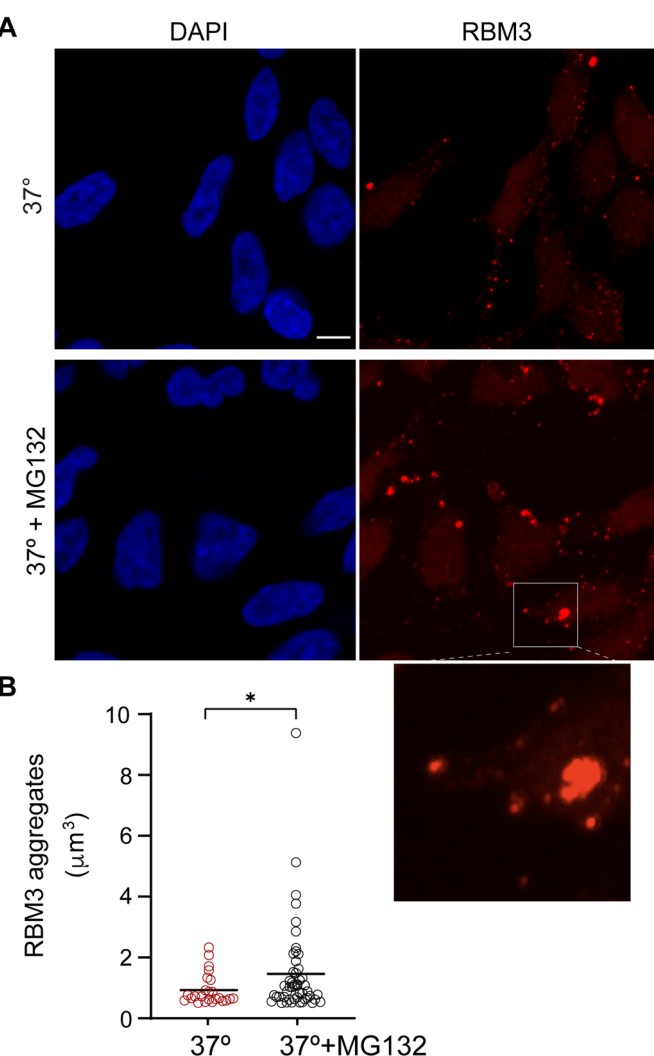

**Fig. 4. RBM3 aggregates form also at 37°C upon proteasome inhibition.** (A) Immunodetection of RBM3 in L-neurons incubated at 37°C and treated with the proteasome inhibitor MG132 at 20 μM for 6 h. The boxed area is magnified below. Scale bar: 5 μm. (B) Quantification of RBM3 aggregates shown in A. The aggregates were scored from circa 260 cells/condition from three independent experiments ($n$=3), with the size cutoff of 0.5 μm³. Aggregate per cell: 0.09 at 37°C and 0.19 upon MG132 treatment. Welch's $t$-test was used for comparison between the two conditions. *$P$=0.02.

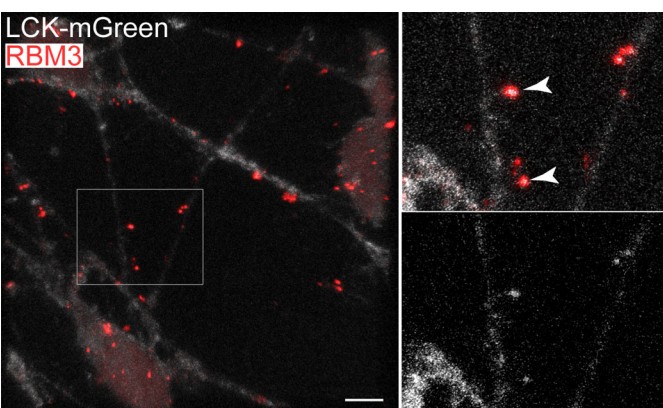

**Fig. 3. Neuronal RBM3 aggregates localize to the cytoplasm in cell soma and neurites.** Immunodetection of RBM3 in L-neurons expressing plasma membrane-targeted LCK-mGreenLantern incubated at 10°C. The boxed area in the left panel is magnified on the right, showing RBM3 and LCK-mGreenLantern (above) or only LCK-mGreenLantern (below). Note that some RBM3 aggregates appear to bulge off from neurites (arrowheads). Scale bar: 5 μm.

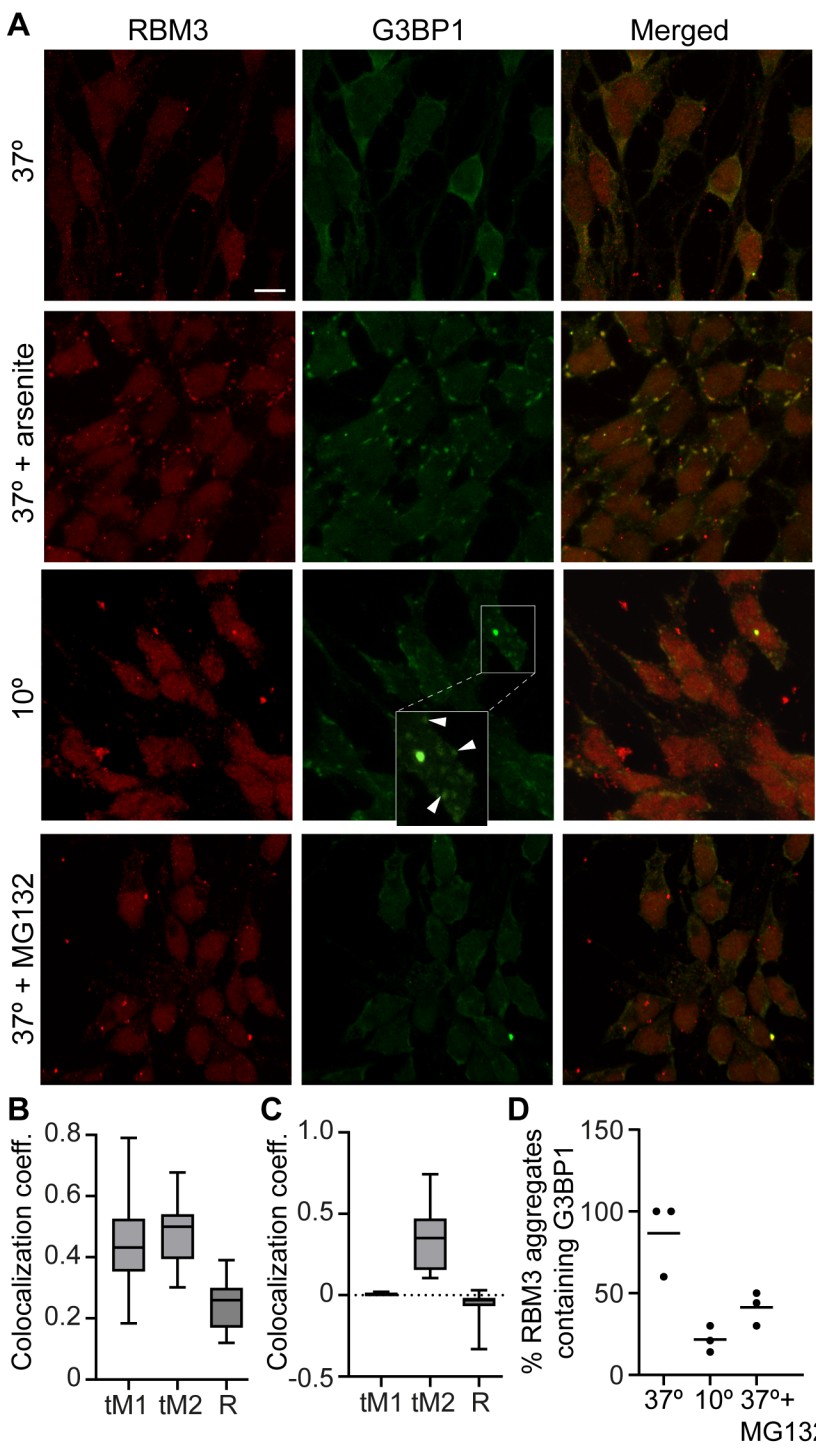

**Fig. 5. RBM3 aggregates are distinct from stress granules.** (A) Double immunofluorescence of RBM3 and G3BP1 (a stress granule marker) in L-neurons treated as labelled. The arsenite was used at 500 µM for 1 h and MG132 at 20 µM for 6 h. The boxed area is shown magnified, the arrow heads point to some of the SGs. Scale bar: 10 µm. (B,C) Colocalization analysis of arsenite and cold-induced stress granules, respectively, with Mander's correlation coefficient (tM1 and tM2) and Pearson's correlation coefficient (R). The value of tM1 is the fraction of RBM3 colocalizing with G3BP1 and tM2 is the fraction of G3BP1 colocalizing with RBM3. (D) The percentage RBM3 aggregates also containing G3BP1 at 37°C, 10°C, or upon MG132 treatment at 37°C. The numbers were scored from around 240 cells/condition from three independent experiments ($n$=3).

did not observe the formation of G3BP1-labelled SGs at 37°C in MG132-treated cells that accumulated RBM3 aggregates (Fig. 5A). Furthermore, only some RBM3 aggregates contained G3BP1 (Fig. 5D). Combined, these observations suggest that RBM3 aggregates are distinct from SGs.

## RBM3 aggregation occurs in different types of neurons

Thus far, we detected RBM3 aggregates in L-neurons. To determine if the aggregation also happens in other types of neurons, we chose the human SH-SY5Y neuroblastoma cell line, which can be differentiated into neuron-like cells (Ross et al., 1983). Like the

L-neurons, we subjected differentiated SH-SY5Y cells to acute cold or MG132 at 37°C and in each case observed the formation of RBM3 aggregates (Fig. 6A,C, and Fig. S5A). However, we noticed that the aggregates were less abundant, possibly reflecting a less complete differentiation of SH-SY5Y cell than LUHMES cells. To test the specificity of RBM3 staining, we knocked out RBM3 in SH-SY5Y and observed the expected loss of RBM3 signal across all experimental conditions (Fig. 6B). Additionally, we overexpressed GFP-tagged RBM3 in SH-SY5Y cells. Consistent with the findings using antibodies, we observed the aggregation of RBM3-GFP following the exposure to either MG132 or cold (Fig. S5B). All

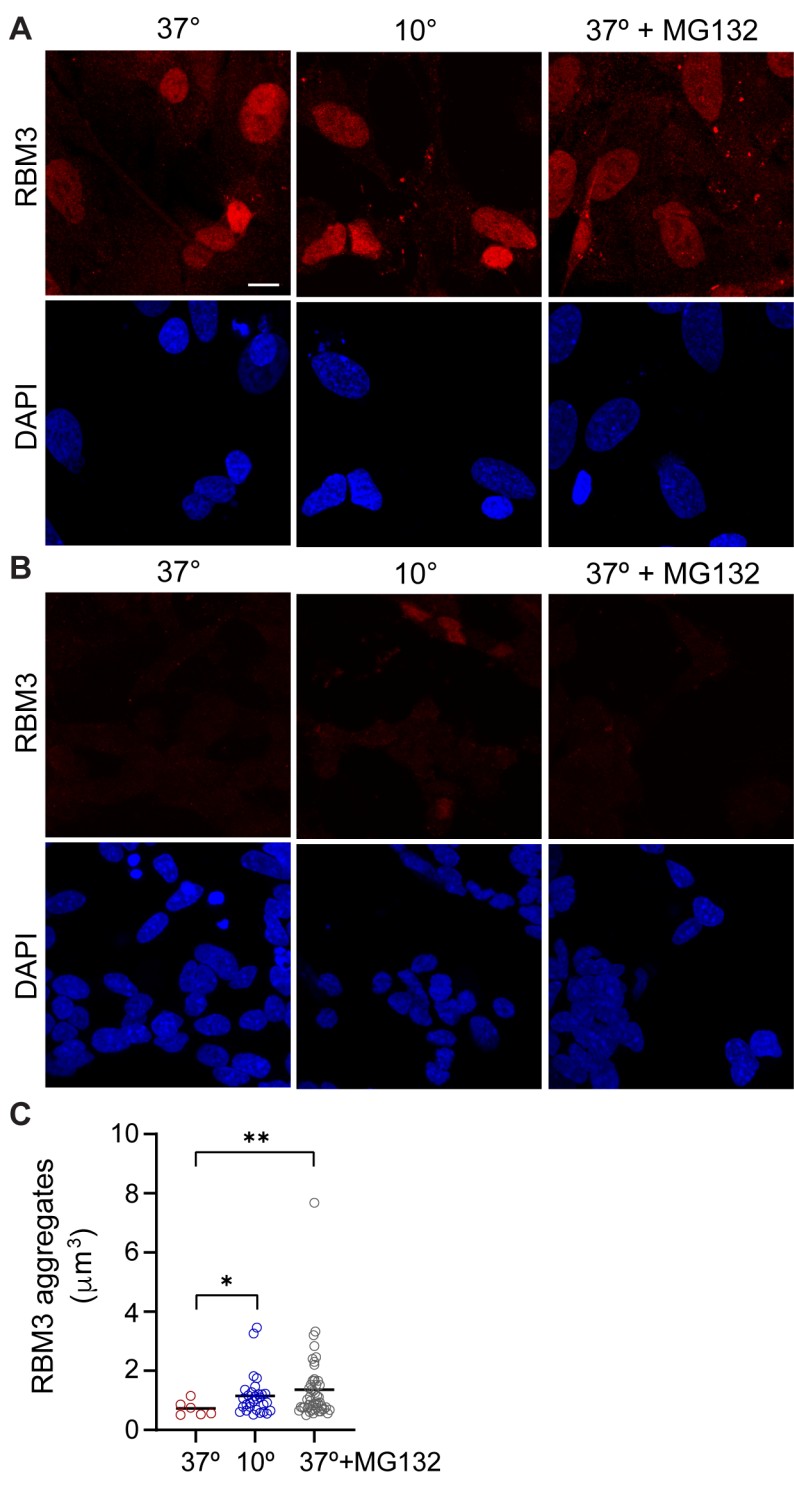

Fig. 6. RBM3 aggregates also form in SH-SY5Y-RBM3 neurons. (A) Immunofluorescence of RBM3 and DAPI-stained wild-type SH-SY5Y cells treated as indicated. MG132 was used at 15 µM for 5 h. Scale bar: 10 µm. (B) Immunofluorescence of RBM3 and DAPI-stained SH-SY5Y-RBM3 KO cell pool with 94% KO efficiency. Magnification as in A. (C) Quantification of RBM3 aggregates in wild-type SH-SY5Y cells. Around 75 cells were scored from three different experiments (*n*=3), with the size cutoff of 0.5 µm³. The number of RBM3 aggregates per cell: 0.08 at 37°C, 0.4 at 10°C and 0.64 upon MG132 treatment. Welch's *t*-test was used for two-condition comparisons. In 37°C versus 10°C, *P=0.015; and in 37°C versus MG132, **P=0.002.

combined, our findings suggest that the human RBM3 has the propensity for aggregation in differentiated neurons. While this aggregation can be induced by acute hypothermia, the aggregation may also happen at a physiological temperature but is normally prevented by a mechanism involving the proteasome.

### RBM3 isoform-specific aggregation?
By contrast to human neurons, murine neurons express two protein isoforms of RBM3 differing by the presence or absence of a specific arginine, corresponding to the Arg at the position 135 in the murine protein (Fig. 7A; for simplicity, we refer to the corresponding Arg as Arg[135] across all species). While the murine Arg[135+] isoform is expressed in the nucleus, soma, and dendritic spines, the Arg[135−] isoform is expressed predominately in the dendrites (Smart et al., 2007). Intriguingly, it is the Arg[135−] isoform that, when overexpressed, was shown to protect neurons from degeneration in disease models (Peretti et al., 2015). Since posttranslational modifications of Arg, most notably methylation, can impact various aspects of protein behaviour, we used NCBI Eukaryotic Genome Annotation Pipeline to examine the distribution of RBM3 isoforms

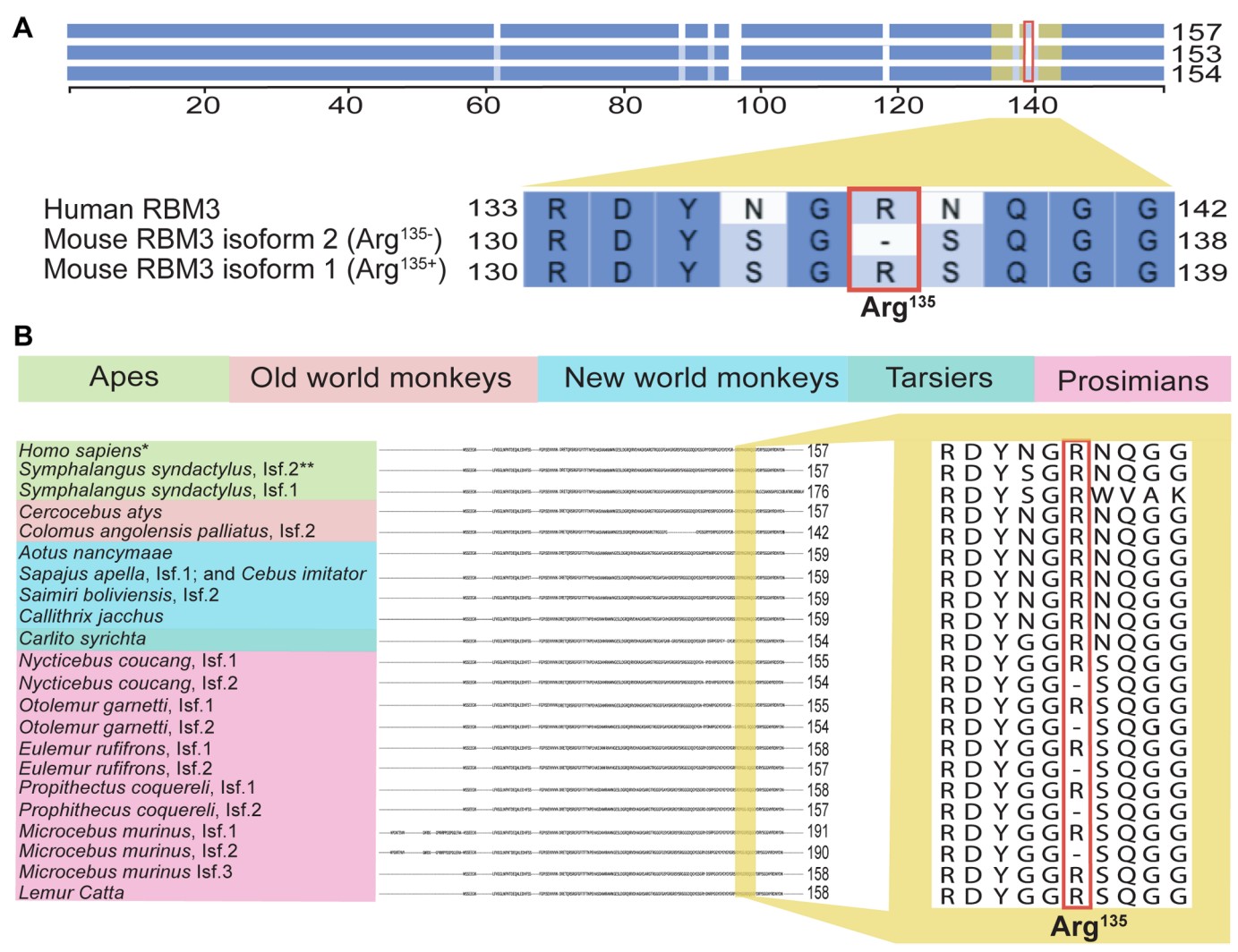

**Fig. 7. Occurrence of Arg$^{135+/-}$ RBM3 isoforms in mammals.** (A) Alignment of human and murine RBM3 proteins. The scale indicates positions of amino acids and the purple shading the amino acid similarity. The yellow region is magnified below. The two murine RBM3 isoforms are identical except for the presence/absence of a single arginine at the position 135 in the isoform 2 (hereby Arg$^{135-}$, boxed in red). (B) Alignment of all primate RBM3 proteins from the NCBI RefSeq database annotated as human RBM3 orthologs. On the left are primate species color-coded based on the groups. If labelled as such in the database, we indicate the RBM3 isoform (Isf.). The region magnified in yellow on the right corresponds to the yellow region in A. *16 additional species (4 apes and 12 old world monkeys) have the identical protein; **2 additional species (apes) have the identical protein. Note that only prosimians – the only group of primates displaying hibernation/torpor – includes species expressing the Arg$^{135-}$ isoform of RBM3.

across primates. In addition to the full-length RBM3, some primates express shorter protein variants missing the C-terminal part containing Arg$^{135}$. Excluding such shorter proteins, only (5/35) of the analysed species express the Arg$^{135-}$ RBM3 (Fig. 7B and Fig. S6). This is in stark contrast with other mammalian orders (including rodents, carnivores, chiroptera, and cetartiodactyla), where the vast majority (>95%) of the examined species express the Arg$^{135-}$ isoform (Fig. S6). Interestingly, all primates expressing the Arg$^{135-}$ isoform belong to the Prosimian family containing lemurs and lorises (Fig. 7B), which are the only primates known to hibernate/display cold-induced torpor (Dausmann et al., 2004; Royo et al., 2019; Ruf et al., 2015; Schmid, 2000). If the exclusion of Arg$^{135}$ facilitates RBM3 function in the cold, the Arg$^{135-}$ isoform should be widespread among hibernators and animals capable of entering cold-induced torpor. To examine this, we searched the literature for such animals and matched them against the species expressing RBM3 orthologs using the NCBI Eukaryotic Genome Annotation Pipeline (see Table S1 and the Materials and Methods for details). Except

for two species (the Etruscan shrew, *Suncus etruscus* and the European hedgehog, *Erinaceus europaeus*), the vast majority (18/21) of examined animals express both Arg$^{135-}$ and Arg$^{135+}$ isoforms (Fig. S7). These observations are consistent with the hypothesis that Arg$^{135-}$ RBM3 could be adapted to functioning at low temperatures, perhaps by impeding its aggregation.

## DISCUSSION
We report here the neuronal aggregation of RBM3 protein depending on temperature or proteasomal inhibition. *In vitro*, the RRM domain of RBM3 was reported to oligomerize via nonspecific interactions, and this oligomerization is favored at low temperature (Roy et al., 2022). Also, the RGG/RG motifs found in RBM3 are often found in intrinsically disordered protein regions (IDRs) that are prone to aggregation and phase separation (Chong et al., 2018; Zeke et al., 2022). Thus, RBM3 may be inherently prone to aggregate. However, its aggregation *in vivo* is also context-dependent, as we observed the RBM3 aggregates in differentiated, rather than undifferentiated

neurons. One possible explanation could be that specific molecular milieu in cold-treated neurons (or neurons with compromised proteasomal function) facilitates the aggregation. For example, cold-mediated depolymerization of axonal and dendritic microtubules will impair microtubule-dependent trafficking causing a 'traffic jam' of transported cargo possibly leading to protein aggregation. An alternative (or additional) explanation is that a putative mechanism preventing the aggregation is less efficient in mature neurons so, when challenged by cold or proteasomal dysfunction, it can no longer prevent the aggregation.

Our observation that RBM3 protein aggregates upon proteasomal inhibition suggests that the proteasome – directly or indirectly – counteracts the aggregation. The aggregation observed in cold-treated neurons could have a similar etiology. However, the absence of RBM3 ubiquitination in cold or MG132-treated neurons may suggest a ubiquitin-independent mechanism. In line with this hypothesis, intrinsically disordered proteins are degraded by the proteasome in ubiquitin-independent fashion (Thomas et al., 2023). Also, a moderate decrease in temperature promotes the ubiquitin-independent proteasomal degradation of disease-related proteins in nematodes and cultured human cells (Lee et al., 2023). Similarly, our former study in nematodes demonstrated the importance of proteasome for surviving severe cold, although whether this function depends on ubiquitin remains to be tested (Habacher et al., 2016). Finally, a recent study from non-neuronal cells suggests that RBM3 may be degraded by the proteasome in ubiquitin-independent manner (Xie et al., 2023). Thus, while the direct evidence is currently missing, multiple lines of evidence suggest that ubiquitin-independent proteasomal degradation may prevent the neuronal aggregation of RBM3. Conversely, pathological or environmental conditions (like hypothermia) compromising proteasomal activity would favor RBM3 aggregation.

Humans and other non-hibernators encounter severe cold potentially resulting in RBM3 aggregation only occasionally, for example in the extremities or across the body during induced or accidental hypothermia. Hibernators, by contrast, experience severe cold routinely as part of the life cycle. If the aggregation results in the loss of RBM3 function, hibernators could have evolved mechanisms preventing the aggregation. Along these lines, we speculate that the $Arg^{135-}$ isoform of RBM3 might be resistant to aggregation. This arginine is found in the C-terminal low complexity fragment of RBM3, and its inclusion and/or posttranslational modification (e.g. via methylation) could impact protein behavior. While testing this hypothesis in hibernators is challenging, the expression of human $Arg^{135-}$ RBM3 variant in cultured neurons should clarify the importance of $Arg^{135}$ for the RBM3 aggregation.

While reporting a novel aspect of RBM3 biology, our study leaves several open questions. For example, we do not currently know if RBM3 aggregates contain additional proteins and/or RNA. Also, the fate of RBM3 aggregates remains unclear. A ubiquitin-dependent degradation of crystallin aggregates during rewarming from hibernation was reported in ground squirrels (Yang et al., 2024). Thus, both ubiquitin dependent and independent activity of the proteasome may be important for the recovery from hypothermia. Finally, we noticed that some RBM3 aggregates appear to be caught in the act of budding off from the neurites. If so, are they taken up by the neighboring glia cells like the *Caenorhabditis elegans* exopheres (Wang et al., 2023b), and do they function as garbage bins or might they have a biological function? Future work will be necessary to address these and other outstanding questions regarding the biology of this important neuronal actor.

## MATERIALS AND METHODS
### Antibodies
The following antibodies were used for immunofluorescence (IF) and immunoblotting (WB): rabbit RBM3 polyclonal antibody [Proteintech, cat. no.: 14363-1-AP, 1:500 (IF) and 1:5000 (WB)], mouse G3BP1 monoclonal antibody [Proteintech, cat. no.: 66486-1-Ig, 1:500 (IF)], mouse actin monoclonal antibody [MilliporeSigma, MAB1501, 1:10,000 (WB)], mouse ubiquitin monoclonal antibody [Abcam, ab303664, 1:2000 (WB)], and Alexa Fluor IgG secondary antibodies [Thermo Fisher Scientific, 1:1500 (IF)].

### Generation of stable cell lines (LUHMES-LCK-mGreenLantern, SH-SY5Y-RBM3-eGFP and SH-SY5Y-RBM3-KO)
Plasmid containing LCK-mGreenLantern was a kind gift from Sverre Grødem (University of Oslo, Norway). The LCK-mGreenLantern sequence was cloned into lentiviral transfer vector, pLJM1-eGFP (plasmid #19319) with the addition of NheI and EcoRI at the 5′ and 3′ end, respectively.

To generate the RBM3-GFP vector, full length RBM3 from pEGFPN1-RBM3 (Addgene, plasmid #87860) was retrieved by cutting with NheI and AgeI at the 5′ and 3′ end, respectively. The cut-out sequence was thereafter inserted into the lentiviral transfer vector, pLJM1-eGFP at the respective restriction sites. Lentiviral particles were created by transfecting the lentiviral transfer plasmid along with packaging plasmids (Lentivirus Packaging Mix, Mirus Bio, cat. no. MIR6630) into HEK293 cells using TransIT-LT1 transfection reagent (Mirus Bio, cat. no. MIR2304) according to the manufacturer's instructions. The media was collected and filtered with a low protein-binding 0.45 µm filter (Millipore, SLHPR33RS) and the lentiviral particles were stored at −80°C. For transduction, lentivirus solution was thawed on ice and added dropwise to 70-80% confluent cells. Transduced cells were selected by the addition of puromycin (6 µg/ml for LUHMES cells and 3 µg/ml for SH-SYFY cells) for 24 h. The selection was repeated after 4 days, and the surviving cells were allowed to expand. The cells were subsequently used as a polyclonal line.

The SH-SY5Y-RBM3-KO cells were generated by Genscript using lentiviral method with RBM3 sgRNA (TCAAGGACCGGGAGACTCAG). RBM3 KO-Score of 94% was determined by sequencing trace analysis with CRISPR analysis tool (CAT) by Genscript. KO was further confirmed by IF (Fig. 6).

### Cell culture, transfections, and stable cell lines
LUHMES cells were cultured according to the culture conditions previously described (Scholz et al., 2011). In brief, LUHMES cells were grown in flasks coated with poly-L-ornithine hydrobromide (Merck, P3655) and human fibronectin (Merck, F1056). The growth media was Adv. DMEM/F12 media (Thermo Fisher Scientific, 12634028) supplemented with L-GlutaMax (Thermo Fisher Scientific, 35050061), N2 supplement (Thermo Fisher Scientific, 17502048), human bFGF (R&D Systems 4114-TC) and 0.5% P/S. To induce differentiation, proliferation media was replaced with Adv. DMEM/F12 media supplemented with 1X L-GlutaMax (Thermo Fisher Scientific, 35050061), 1X N-2 supplement (Thermo Fisher Scientific, 17502048), GDNF (20 µg/ml), 100 mM cAMP (Merck, D0627) and 1 mg/ml tetracycline (Merck, T8032) and allowed to differentiate for 5 days.

The SH-SY5Y cells were cultured according to the culture conditions previously described (Shipley et al., 2016). In brief, cells were cultured in DMEM/F12 media (Thermo Fisher Scientific, 11320033) supplemented with 10% fetal bovine serum (FBS) and 0.5% P/S. To induce differentiation, DMEM/F12 media was supplemented with 1% FBS and 10 µM retinoic acid (MCE, HY-14649) and allowed to differentiate for 12-14 days.

### Treatment conditions
Cold stress was induced by placing culture plates in a cell culture incubator maintained at 10°C with 5% $CO_2$. For rewarming, cells were taken out from the cold incubator and directly placed in a 37°C incubator with 5% $CO_2$. Sodium arsenite (Merck, 106277) was used at a concentration of 500 µM for 1 h. MG132 (MCE, HY-13259) was used at a concentration of 15-20 µM for 5-6 h.

## Protein extraction and western blotting

After treatments, cells were lysed with RIPA buffer [50 mM Tris HCl, pH 7.4, 150 mM NaCl, 1% Triton X-100, 0.5% sodium deoxycholate, 0.1% SDS along with complete protease inhibitor cocktail (Merck, 11836153001)].

Protein concentration was measured by Bradford reagent (Merck, B6916). The cell lysates were resolved by SDS-PAGE (NuPAGE Bis-Tris precast polyacrylamide gels, Thermo Fisher Scientific, NP0321) and transferred to Immobilon-FL PVDF membrane (Millipore, IPFL00010). After transfer, membranes were rinsed with 1X TBS and 1X TBS-T (TBS-Tween20) and then incubated in blocking buffer (Intercept Blocking Buffer, LICORbio, 927-70001) for 1 h at room temperature (RT). The membranes were incubated overnight in primary antibodies, washed 3X TBS-T and then incubated in IRDye secondary antibodies (LICORbio, 1:10000 dilution) for 1 h at RT. The blots were rinsed in 3X TBS, air dried and scanned with Odyssey CLX (LICORbio).

## Immunofluorescence

LUHMES cells were cultured on coated coverslips (PLO and fibronectin). SH-SY5Y cells were grown on laminin (1.5 µg/ml) coated coverslips. After treatments, cells were fixed in 4% PFA (Thermo Fisher Scientific, 28906), washed 3×5 min with PBS. Cells were washed (2×1 min) with wash buffer (0.1% BSA in PBS) before permeabilizing with 0.3% TritonX-100 in PBS for 10 min. Cells were then blocked in 5% goat serum for 30 min before adding primary antibodies in blocking solution and incubated overnight at 4°C. Following primary antibody incubation, the cells were rinsed 3×5 min in wash buffer and incubated in secondary antibodies for 1 h at RT. After incubation, the cells were washed 3×5 min in wash buffer and mounted in Mowiol (Sigma-Aldrich, cat. no. 81381) and stored at 4°C until visualized with a fluorescence microscope.

## Imaging and data analysis

Confocal Z-stack images were acquired with a Zeiss LSM 880 microscope. Images were analyzed in FIJI (ImageJ) or Imaris. Quantification of RBM3 aggregates were performed using Imaris (v.9.5.1). Equal number of cells from each condition were used for quantification (75-300 cells from at least three biological replicates were used for analysis). RBM3 aggregates of volume lower than 0.5 µm$^3$ were arbitrarily excluded from the analysis in the main figures and 0.3 µm$^3$ cut off (closer to the resolving capacity of a confocal microscope) was applied in the supplementary images.

Colocalization analysis of RBM3 and G3BP1 within SGs was performed in ImageJ using the Coloc2 plugin. Maximum-intensity projection images of substacks containing SGs were used. After background subtraction, the ROIs were manually selected and analyzed with Coloc2 to obtain Mander's coefficients for each channel (tM1 and tM2) and the Pearson's correlation coefficient (R). tM1 represents the fraction of RBM3 colocalizing with G3BP1 and tM2 is the fraction of G3BP1 colocalizing with RBM3.

All statistical analyses were performed using the Graphpad Prism 10 software.

## Alignment of sequenced RBM3 protein

RBM3 orthologs were identified using NCBI Eukaryotic Genome Annotation Pipeline (Thibaud-Nissen et al., 2016). To align the proteins, we used different approaches. As a starting point, sequences of all proteins were downloaded (.FASTA) and aligned using COBALT (Papadopoulos and Agarwala, 2007). The aligned C-termini were used to make a dataset where the amino acid at Arg$^{135}$ for every analyzed protein was identified. Additional information such as order, family, and common names was added to the dataset. Based on collected information, the number of species expressing the Arg$^{135−}$ isoforms was calculated for each animal order. To visualize the aligned RBM3 proteins across primates, the UniProt Alignment tool was used (Ahmad et al., 2025). The selection of hibernators was based on Geiser (2013); this selection was matched against the available RBM3 orthologs within the dataset (Table S1).

## Acknowledgements

We thank Sverre Grødem for discussions and help with transgenic cells, Sverre Grødem and Pooja Kumari for comments on the manuscript, and the Oslo NorMIC Imaging Platform at the Department of Biosciences for supporting this work.

The SH-SY5Y cell line was a kind gift from Prof. Ragnhild E Paulsen, Department of Pharmacy, University of Oslo.

## Competing interests

The authors declare no competing or financial interests.

## Author contributions

Conceptualization: R.C.; Formal analysis: S.K., T.K.; Funding acquisition: R.C.; Investigation: S.K., T.K., R.C.; Methodology: S.K., T.K., R.C.; Project administration: R.C.; Resources: R.C.; Supervision: R.C.; Validation: S.K.; Visualization: S.K., T.K.; Writing – original draft: S.K., T.K., R.C.

## Funding

Some research leading to these results received funding from the Norwegian Financial Mechanism 2014–2021 operated by the Polish National Science Center under the project contract no. UMO-2019/34/H/NZ3/00691. Open Access funding provided by University of Oslo. Deposited in PMC for immediate release.

## Data and resource availability

All relevant data and details of resources can be found within the article and its supplementary information.

## Peer review history

The peer review history is available online at https://journals.biologists.com/bio/lookup/doi/10.1242/bio.062179.reviewer-comments.pdf

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
