## [Peer Review File · Biology Open]

Human RBM3 protein is prone to form neuronal aggregates opposed by the proteasome

Suman Kumar, Tina Kleven and Rafal Ciosk

DOI: 10.1242/bio.062179

Editor: Catherine L. Jackson

Review timeline

Original submission:	24 July 2025
Editorial decision:	28 July 2025
First revision received:	28 November 2025
Accepted:	8 December 2025

Original submission

First decision letter

MS ID#: bio.062179

MS Title: Human RBM3 protein is prone to form neuronal aggregates opposed by the proteasome

Authors: Suman Kumar, Tina Kleven and Rafal Ciosk

Article Type: Research Article

I have now reached a decision on the above manuscript.

The reviewer reports are shown at the bottom of this email or can be accessed, together with a copy of this decision letter, by going to:

As you will see, the reviewers raised a number of substantial criticisms that prevent me from accepting the paper at this stage.

They suggest, however, that a revised version might prove acceptable, if you can address their concerns. If you think that you can deal satisfactorily with the criticisms on revision, I would be pleased to see a revised manuscript. We would then return it to the reviewers. At this stage, we also ask you to ensure your manuscript complies with our formatting guidelines. Provided you are able to fully address the referees' comments, we are positive about publication of your paper (we accept over 95% of revision submissions) and therefore hope you won't mind any extra work involved in reformatting your manuscript at this point.

Please ensure that you clearly highlight all changes made in the revised manuscript. Please avoid using 'Tracked changes' in Word files as these are lost in PDF conversion.

I should be grateful if you would also provide a point-by-point response detailing how you have dealt with the points raised by the reviewers in the 'Response to Reviewers' box. Please attend to all of the reviewers' comments. If you do not agree with any of their criticisms or suggestions please explain clearly why this is so.

Reviewer 1

Comments for the author

Kumar et al. explores the aggregation behavior of the human RBM3 protein under hypothermic stress and proteasomal inhibition in neuron-like cells. While the topic is potentially interesting and timely, the experimental evidence presented is too preliminary to support the major conclusions of the paper. Many results remain descriptive, and key controls and quantification are missing. Additional mechanistic experiments would substantially strengthen the findings. Therefore, in my opinion, the experimental evidences are insufficient to justify publication at this stage. I recommend major revision with substantial new experiments and quantification:

- 1) RBM3 is also present in small aggregates at 37°C, although to a lesser extent than at 10°C. The authors should properly quantify this phenotype. Moreover, without a cytosolic marker or a brightfield image, it is impossible to determine whether these aggregates are located inside or outside the neurons. Additionally, an RBM3 knockdown experiment to verify the specificity of the primary antibody would significantly strengthen the conclusions.
- 2) The authors should specify the duration of neuronal exposure to mild or acute hypothermia in Fig. 1.
- 3) Aggregate size cut-offs are arbitrary and not fully justified in the text.
- 4) The RBM3 protein levels shown in Fig. 1B should be properly analyzed by repeating the experiment with biological replicates and including appropriate statistical analysis. Reporting data from a single experiment (N=1) is not acceptable. For publication-ready study, at least 3 independent biological replicates should be performed.
- 5) The authors should experimentally demonstrate that the cooling procedures used in this study do not impair neuronal viability by performing an appropriate viability assay. This is particularly important in light of the representative data shown in Fig. 1C, where the nuclei of neurons exposed to 10°C appear morphologically altered compared to those at 37°C. The presence of small DAPI-positive puncta may indicate neuronal death.
- 6) The authors should include representative images and quantification of RBM3 localization at 30°C in Fig. 1C, as referenced in the text.
- 7) While the figure legends mention the number of aggregates, the data presentation is often limited (e.g. number of aggregates per 200 cells rather than per cell, lack variation/SD in aggregate size).
- 8) Statistical analysis should be performed in Fig. 1D. I also recommend that the authors use a brightfield image or a cytosolic marker to quantify the number of aggregates per cell.
- 9) The results shown in Fig. 2 are purely observational. Without proper quantification, this figure adds little value. Furthermore, the authors should co-stain with ubiquitin to determine whether the RBM3 aggregates are ubiquitinated. The data from purified proteins currently included are insufficient to support any conclusions.
- 10) Figure 4A lacks a critical control: neurons maintained at 37°C should be included.
- 11) The RBM3 aggregates shown in Fig. 5 should be quantified to allow comparison between experimental conditions.
- 12) The hypothesis that Arg135- and Arg135+ isoforms differentially influence RBM3 aggregation should be experimentally tested.

Reviewer 2

Comments for the author

This paper investigates the aggregation of human RBM3 protein in neurons under cold temperatures (10 °C) and following proteasome inhibition. RBM3 is an RNA-binding protein whose expression is induced by moderately low temperatures (e.g., 30 °C) in mammals and is implicated in various neuroprotective functions. However, how RBM3 behaves at lower temperatures—such as those used in extreme medical hypothermia (10 °C)—remains poorly understood. This is an important question, as such low temperatures are clinically relevant in specific emergency treatments. Therefore, the study has the potential to provide meaningful insights into the molecular mechanisms of neuronal responses to hypothermia.

While the topic is important and the study addresses an underexplored area, there are several technical concerns that need to be addressed.

Major Concerns

1. The authors conclude that most RBM3 aggregates are distinct from stress granules (SGs), based on the results shown in Fig 4. However, the evidence presented is unconvincing.

First, the claim that "RBM3 was predominantly restricted to larger aggregates, with no detectable RBM3 signal in the smaller SG (Fig. 4A). Also, the majority (circa 75%) of RBM3 aggregates did not co-localize with G3BP1 (Fig. 4A-B)." is not well supported. Magnified views should be provided so readers can assess aggregate sizes. Quantification alone is not sufficient; statistical analysis is needed to support these claims.

Second, in Fig 4C, SYTO Green was used to label SGs. However, the images show little to no SGs—SYTO Green typically labels cytoplasmic RNA granules, but its absence here raises questions about the method's validity in this context. The authors should clarify this issue or use additional SG markers to validate their conclusion.

2. The authors report that RBM3 aggregates form in the cytoplasm of both neuronal somas and neurites (Figs. 2 and 3A; also supported by Fig. 4C). However, Fig 5 shows RBM3-GFP aggregates predominantly in the nucleus of SH-SY5Y cells. This discrepancy raises questions about whether the observed aggregation behavior is generalizable across cell types. If the faint green regions in Fig 5B represent cytoplasm, a nuclear stain should be included to clearly distinguish nuclear versus cytoplasmic localization.

3. The authors report that RBM3 aggregates are found in dopaminergic-like neurons differentiated from LUHMES cells, but not in undifferentiated LUHMES cells, and conclude that aggregation occurs specifically in mature neurons. However, mature neurons should be defined by more than just differentiation—they should exhibit functional synapses. Without evidence that RBM3 aggregates preferentially form in neurons with synapses, the claim should be softened to state that aggregation occurs in differentiated rather than mature neurons.

4. The sequence comparisons indicating that the absence of Arg135 is common among hibernating or torpor-prone species is intriguing. The authors propose that RBM3 lacking Arg135 may be better adapted to low temperatures, possibly by reducing aggregation. Given the experimental system used in this paper, this hypothesis is readily testable. Testing it would significantly enhance the impact and significance of the study and may make the paper more suitable for publication in Biology Open.

5. Several important experimental results are mentioned without data. For example: Page 5, lines 53-56: "In a preliminary experiment, we purified RBM3 from cold-treated L-neurons to examine by western blotting its potential ubiquitination but found no evidence for it (data not shown)." These data should be shown, even if in supplementary materials. Avoiding "data not shown" improves transparency and strengthens the credibility of the findings.

Minor Concerns

* Page 5, line 50: The reference to "Fig. 3B" should be corrected to "Fig. 3C."

Experimental quality

Does each figure have the proper controls?

If 'No', please indicate reasons in Comments for Author box below.

Reviewer #1:

- No

Reviewer #2:

© 2025. Published by The Company of Biologists under the terms of the Creative Commons Attribution License (<https://creativecommons.org/licenses/by/4.0/>).

- No

Were the data analyzed using appropriate statistical tests?

If 'No', please indicate reasons in Comments for Author box below.

Reviewer #1:

- No

Reviewer #2:

- No

Reproducibility

Were experiments performed using adequate number of biological replicates?

If 'No', please indicate reasons in Comments for Author box below.

Reviewer #1:

- No

Reviewer #2:

- Yes

Does the methods section provide sufficient detail to permit reproducibility?

If 'No', please indicate reasons in Comments for Author box below.

Reviewer #1:

- No

Reviewer #2:

- Yes

Completeness

Are the manuscript's conclusions supported by the data?

If 'No', please indicate reasons in Comments for Author box below.

Reviewer #1:

- No

Reviewer #2:

- No

Scholarship

Do the authors cite and discuss the merits of data that would argue for and against their conclusion?

If 'No', please indicate reasons in Comments for Author box below.

Reviewer #1:

- Yes

Reviewer #2:

- Yes

Does the manuscript title & abstract accurately reflect the contents of the manuscript, without hyperbole?

If 'No', please indicate reasons in Comments for Author box below.

Reviewer #1:

- Yes

Reviewer #2:

- Yes

First revision

Author response to reviewers' comments

Point-by-point response to the reviewer comments:

Reviewer 1:

Kumar et al. explores the aggregation behavior of the human RBM3 protein under hypothermic stress and proteasomal inhibition in neuron-like cells. While the topic is potentially interesting and timely, the experimental evidence presented is too preliminary to support the major conclusions of the paper. Many results remain descriptive, and key controls and quantification are missing. Additional mechanistic experiments would substantially strengthen the findings. Therefore, in my opinion, the experimental evidences are insufficient to justify publication at this stage. I recommend major revision with substantial new experiments and quantification:

We are happy that the reviewer found the topic interesting and timely. Most concerns have been addressed with new experiments and quantifications as explained below. We believe that descriptive studies are within the scope of the journal.

1) RBM3 is also present in small aggregates at 37°C, although to a lesser extent than at 10°C. The authors should properly quantify this phenotype. Moreover, without a cytosolic marker or a brightfield image, it is impossible to determine whether these aggregates are located inside or outside the neurons. Additionally, an RBM3 knockdown experiment to verify the specificity of the primary antibody would significantly strengthen the conclusions.

As requested, we now provided statistical analysis of the aggregates in the new Figs. 2, 4, 6, S2, S4, and S5.

Our microscopy setup does not allow combining brightfield and confocal imaging. However, we examined RBM3 aggregates against the plasma membrane marker LCK-GFP which delineates the whole neuron (soma+neurites). With this marker, we confirmed that the aggregates found away from the soma co-localize with the LCK-delineated neurites (now in Fig. 3).

To address the specificity of RBM3 antibody, we previously GFP-tagged RBM3 in SH-SY5Y cells. Nonetheless, following reviewer's recommendation, we now included an additional experiment, where RBM3 was knocked out in SH-SY5Y cells and the staining confirmed the specificity of antibody staining (new Fig. 6).

2) The authors should specify the duration of neuronal exposure to mild or acute hypothermia in Fig.1.

We now included this information in the figure legend (to Fig. 2).

3) Aggregate size cut-offs are arbitrary and not fully justified in the text.

We now provided an additional graph with the size cutoff just above the resolution limit, thus including all smaller granules (new Fig. S2B). Since the small granules are most abundant, the mean sizes are comparable across the temperatures, despite the obvious emergence of large aggregates in the cold (Fig. S2B). Therefore, to quantify the latter, we introduced an arbitrary cutoff of $0.5 \mu\text{m}^3$ and observed their significant enrichment at 10°C (now in Fig. 2B). We hope that the new figures and the corresponding explanations in the manuscript and legends deliver a clearer view of the RBM3 aggregation behaviour.

4) The RBM3 protein levels shown in Fig. 1B should be properly analyzed by repeating the experiment with biological replicates and including appropriate statistical analysis. Reporting data from a single experiment (N=1) is not acceptable. For publication-ready study, at least 3 independent biological replicates should be performed.

As requested, we have performed two additional experiments, and the corresponding quantification of western blots is now shown in Fig. 1C.

5) The authors should experimentally demonstrate that the cooling procedures used in this study do not impair neuronal viability by performing an appropriate viability assay. This is particularly important in light of the representative data shown in Fig. 1C, where the nuclei of neurons exposed to 10°C appear morphologically altered compared to those at 37°C . The presence of small DAPI-positive puncta may indicate neuronal death.

As shown now in Fig. S1A, exposing L-neurons to hypothermia did not affect their viability. Additionally, we now show in Fig. S1B that cooling leads to a reversible thinning of neurites, which adopt the characteristic “beads-on-string” appearance. This appearance is reversed upon rewarming, consistent with neuronal recovery.

6) The authors should include representative images and quantification of RBM3 localization at 30°C in Fig. 1C, as referenced in the text.

We included them in the new Fig. 2.

7) While the figure legends mention the number of aggregates, the data presentation is often limited (e.g. number of aggregates per 200 cells rather than per cell, lack variation/SD in aggregate size).

As suggested, we included in the legends numbers/cell and performed statistics on the aggregate size distribution.

8) Statistical analysis should be performed in Fig. 1D. I also recommend that the authors use a brightfield image or a cytosolic marker to quantify the number of aggregates per cell.

The statistical analysis is now included in the new Figs. 2, 4, 6, S2, S4, and S5. The aggregates are not visible using brightfield imaging.

9) The results shown in Fig. 2 are purely observational. Without proper quantification, this figure adds little value. Furthermore, the authors should co-stain with ubiquitin to determine whether the RBM3 aggregates are ubiquitinated. The data from purified proteins currently included are insufficient to support any conclusions.

We believe that showing RBM3 staining against LCK-delineated neurons is useful, as it clarifies the association of RBM3 aggregates with neurites. Thus, we decided to show a representative image in Fig. 3 but moved the image rotations (previously in panel B) to the supplemental figure S3.

Alas, the primary antibody against ubiquitin that we used for western blotting turned out to

be unsuitable for immunofluorescence, and we had no time for further troubleshooting.

The western blot experiment is now shown fully (including IP, as also requested by the second reviewer) in the new Fig. S4.

10) Figure 4A lacks a critical control: neurons maintained at 37°C should be included.

We now included the 37°C control and modified the figure to address the comments from reviewer 2 (new Fig. 5).

11) The RBM3 aggregates shown in Fig. 5 should be quantified to allow comparison between experimental conditions.

We modified this figure to include the RBM3 KO cells and show the corresponding quantifications (new Fig. 6) and moved the images of RBM-GFP to the supplemental figure S5.

12) The hypothesis that Arg135⁻ and Arg135⁺ isoforms differentially influence RBM3 aggregation should be experimentally tested.

We attempted to create a cell line expressing the Arg135⁻ isoform but were unable to complete this work since the employment of the first author run out. Nonetheless, we believe the hypothesis gives an exciting evolutionary perspective and formulating it was not trivial, as it required extensive analysis of RBM3 isoforms across many animal groups. We plea that the reviewer agrees to leaving the hypothesis open to future studies.

Reviewer 2:

This paper investigates the aggregation of human RBM3 protein in neurons under cold temperatures (10 °C) and following proteasome inhibition. RBM3 is an RNA-binding protein whose expression is induced by moderately low temperatures (e.g., 30 °C) in mammals and is implicated in various neuroprotective functions. However, how RBM3 behaves at lower temperatures—such as those used in extreme medical hypothermia (10 °C)—remains poorly understood. This is an important question, as such low temperatures are clinically relevant in specific emergency treatments. Therefore, the study has the potential to provide meaningful insights into the molecular mechanisms of neuronal responses to hypothermia. While the topic is important and the study addresses an underexplored area, there are several technical concerns that need to be addressed.

We are happy that the reviewer found the topic important. We addressed most concerns with new experiments and quantifications as explained below.

Major Concerns

1. The authors conclude that most RBM3 aggregates are distinct from stress granules (SGs), based on the results shown in Fig 4. However, the evidence presented is unconvincing.

First, the claim that "RBM3 was predominantly restricted to larger aggregates, with no detectable RBM3 signal in the smaller SG (Fig. 4A). Also, the majority (circa 75%) of RBM3 aggregates did not co-localize with G3BP1 (Fig. 4A-B)." is not well supported. Magnified views should be provided so readers can assess aggregate sizes. Quantification alone is not sufficient; statistical analysis is needed to support these claims.

Second, in Fig 4C, SYTO Green was used to label SGs. However, the images show little to no SGs—SYTO Green typically labels cytoplasmic RNA granules, but its absence here raises questions about the method's validity in this context. The authors should clarify this issue or use additional SG markers to validate their conclusion.

To address the relationship between RBM3 and SGs, we now included a positive control, were cells grown at 37°C were treated with arsenite, which is known to induce the SGs. Under these conditions, we observed a strong co-localization between RBM3 and G3BP1 (new Fig. 5A-B), which agrees with previous reports (referenced in the manuscript). By contrast, the two proteins did not colocalize in the cold, which is now supported by the statistical analysis (Fig. 5A, C). Moreover, while the SGs were present at 10°C, they were absent from MG132 treated cells at 37°C (Fig. 5A), further supporting the notion that RBM3 aggregates are distinct from SGs. Nonetheless, some RBM3 aggregates contain G3BP1 and, to better illustrate this, we replaced the former graph with a new one (Fig. 5D). As for the SYTO staining, the lack of SYTO staining in the aggregates could indeed reflect a technical issue, so we decided to remove this data.

2. The authors report that RBM3 aggregates form in the cytoplasm of both neuronal somas and neurites (Figs. 2 and 3A; also supported by Fig. 4C). However, Fig 5 shows RBM3-GFP aggregates predominantly in the nucleus of SH-SY5Y cells. This discrepancy raises questions about whether the observed aggregation behavior is generalizable across cell types. If the faint green regions in Fig 5B represent cytoplasm, a nuclear stain should be included to clearly distinguish nuclear versus cytoplasmic localization.

We agree that the choice of pictures could give that impression. We now repeated this experiment to include nuclear staining (DAPI) and show the aggregates away from the cell body in the new Fig. S5.

3. The authors report that RBM3 aggregates are found in dopaminergic-like neurons differentiated from LUHMES cells, but not in undifferentiated LUHMES cells, and conclude that aggregation occurs specifically in mature neurons. However, mature neurons should be defined by more than just differentiation—they should exhibit functional synapses. Without evidence that RBM3 aggregates preferentially form in neurons with synapses, the claim should be softened to state that aggregation occurs in differentiated rather than mature neurons.

We thank the reviewer for stressing the importance of distinguishing between differentiated and mature neurons. We have revised the text accordingly.

4. The sequence comparisons indicating that the absence of Arg135 is common among hibernating or torpor-prone species is intriguing. The authors propose that RBM3 lacking Arg135 may be better adapted to low temperatures, possibly by reducing aggregation. Given the experimental system used in this paper, this hypothesis is readily testable. Testing it would significantly enhance the impact and significance of the study and may make the paper more suitable for publication in Biology Open.

We attempted to create a cell line expressing the Arg135- isoform but were unable to complete this work since the employment of the first author run out. Nonetheless, we believe the hypothesis gives an exciting evolutionary perspective and formulating it was not trivial, as it required extensive analysis of RBM3 isoforms across many animal groups. We plea that the reviewer agrees to leaving the hypothesis open to future studies.

5. Several important experimental results are mentioned without data. For example: Page 5, lines 53-56: "In a preliminary experiment, we purified RBM3 from cold-treated L-neurons to examine by western blotting its potential ubiquitination but found no evidence for it (data not shown)." These data should be shown, even if in supplementary materials. Avoiding "data not shown" improves transparency and strengthens the credibility of the findings.

We now show the whole experiment in the supplementary figure S4.

Minor Concerns

** Page 5, line 50: The reference to "Fig. 3B" should be corrected to "Fig. 3C."*

We verified that the figures are now correctly referenced throughout the revised manuscript.

Second decision letter

MS ID#: bio.062179R1

MS Title: Human RBM3 protein is prone to form neuronal aggregates opposed by the proteasome

Authors: Suman Kumar, Tina Kleven and Rafal Ciosk

Article Type: Research Article

I am happy to tell you that your manuscript has been accepted for publication in Biology Open, pending our standard publication integrity checks. It was accepted on 8th December 2025.